# HIV-1–Host Interaction in Gut-Associated Lymphoid Tissue (GALT): Effects on Local Environment and Comorbidities

**DOI:** 10.3390/ijms241512193

**Published:** 2023-07-30

**Authors:** Sonia Moretti, Ivan Schietroma, Giuseppe Sberna, Maria Teresa Maggiorella, Leonardo Sernicola, Stefania Farcomeni, Marta Giovanetti, Massimo Ciccozzi, Alessandra Borsetti

**Affiliations:** 1National HIV/AIDS Research Center, Istituto Superiore di Sanità, 00162 Rome, Italy; sonia.moretti@iss.it (S.M.); ivan.schietroma@iss.it (I.S.); giuseppe.sberna@iss.it (G.S.); mariateresa.maggiorella@iss.it (M.T.M.); leonardo.sernicola@iss.it (L.S.); stefania.farcomeni@iss.it (S.F.); 2Instituto Rene Rachou, Fundação Oswaldo Cruz, Belo Horizonte 30190-009, Minas Gerais, Brazil; giovanetti.marta@gmail.com; 3Sciences and Technologies for Sustainable Development and One Health, University Campus Bio-Medico of Rome, 00128 Rome, Italy; 4Unit of Medical Statistics and Molecular Epidemiology, University Campus Bio-Medico of Rome, 00128 Rome, Italy; m.ciccozzi@unicampus.it

**Keywords:** HIV-1 pathogenesis, GALT, gut dysbiosis, Th17, Treg, Tfh, HIV-1 reservoir, comorbidities

## Abstract

HIV-1 replication in the gastrointestinal (GI) tract causes severe CD4+ T-cell depletion and disruption of the protective epithelial barrier in the intestinal mucosa, causing microbial translocation, the main driver of inflammation and immune activation, even in people living with HIV (PLWH) taking antiretroviral drug therapy. The higher levels of HIV DNA in the gut compared to the blood highlight the importance of the gut as a viral reservoir. CD4+ T-cell subsets in the gut differ in phenotypic characteristics and differentiation status from the ones in other tissues or in peripheral blood, and little is still known about the mechanisms by which the persistence of HIV is maintained at this anatomical site. This review aims to describe the interaction with key subsets of CD4+ T cells in the intestinal mucosa targeted by HIV-1 and the role of gut microbiome and its metabolites in HIV-associated systemic inflammation and immune activation that are crucial in the pathogenesis of HIV infection and related comorbidities.

## 1. Introduction

Active antiretroviral therapy (ART) effectively suppresses viral replication in people living with HIV (PLWH), reducing viremia to undetectable levels. Moreover, ART plays a crucial role in partially restoring the immune system’s functionality and significantly slowing down the progression of AIDS [1]. During the initial 1–4 years of ART, the number of CD4+ T cells harbouring HIV DNA in the blood declines and then plateaus, resulting in a stable reservoir of infected cells that may contribute to possible persistent residual viral replication [2,3]. Indeed, due to the poor penetration of the drugs, HIV persists within the organism in particular districts, both cellular and tissue, called sanctuaries that function as true reservoirs where the virus, upon reactivation, is able to replicate and thus represent the major obstacle to eradication of the infection [4,5,6]. In these sanctuaries, HIV can exist in the form of a latent provirus invisible to drugs and to the immune system, so therapy leads to chronicity of the infection but not to eradication of the virus. Treatment interruption causes a rebound in viremia because of the existence of latent long-lived viral reservoirs established in the early stage of infection [7]. Different cell types are considered important reservoirs of HIV latency, including macrophages, dendritic cells (DC) and microglial cells, but resting memory CD4+ T cells constitute the major reservoir of latent provirus for the long-term maintenance of the viral genome [8,9,10]. These long-lived cells are able to proliferate homeostatically, i.e., in the absence of antigenic stimulation, providing “ad aeternum” persistence of the infection, as evidenced by the presence of HIV-infected CD4+ T-cell clones of about 18 years old. In addition to CD4+ T lymphocytes, myeloid cells and, in particular, tissue-resident macrophages are also considered part of the therapy-resistant viral reservoir [9,10]. There are also numerous tissue compartments that serve as reservoirs for the virus that are formed from the earliest stages of infection, and these are represented by the lymphoid organs, the central nervous system (CNS), the gastrointestinal tract (GI) and the gonads, in which the “latent virus reservoir” can reactivate periodically and begin to replicate and infect new cells [11,12,13].

In PLWH, CD4+ memory T cells in the gut carry higher levels of HIV DNA compared to those present in the blood, pointing out the importance of this compartment as a viral reservoir [14]. The gut-associated lymphoid tissue (GALT) is an important part of the immune system, and HIV-induced disruption of the GALT leads to inflammation and immune activation, which may persist even for years [15,16]. Emerging research highlights the importance of the gut microbiome and its metabolites in HIV-infected associated systemic inflammation and immune activation [17,18,19]. Changes in the gut caused by the virus, such as dysbiosis and microbial translocation, contribute to inflammation and immune activation that are crucial in the pathogenesis of HIV infection [17,18,19].

## 2. The Gastrointestinal Tract as a Primary Site of HIV Early Replication and Reservoir

### 2.1. The Gut Barrier Function

The surface of the intestinal mucosa is lined by the epithelium and mucus layers that play an important role in the protection of the human body to prevent contact between the internal and external environments as a physical barrier to commensal and pathogenic microorganisms, toxins and microbial products and/or allergens [20]. The intestinal barrier has different levels of defence mechanisms to limit the translocation of antigens out of the lumen. In healthy individuals, the intestinal barrier system exists as a layer of cells of which the enterocytes constitute the major type and depends on interactions among intercellular tight junctions, antimicrobial peptides secreted by Paneth cells, an adhesive mucous gel layer secreted by Goblet cells and neuroendocrine cells that release intestinal hormones [21,22]. The intestinal barrier is marked by junctional complexes such as (i) tight junctions that comprise transmembrane proteins such as claudin, occludin and junctional adhesion molecule-A (JAM-A) that form cellular channels to regulate paracellular passage of molecules, (ii) adherens junctions that consist of the transmembrane protein E-cadherin and catenin complexes and (iii) highly specialised anchoring junctions called desmosomes that provide cell–cell adhesion via the cadherins desmoglein (Dsg)2 and desmocollin (Dsc)2. In Paneth cells, pattern recognition receptors (PRRs) play a critical role in the penetration of microorganisms, secretion of mucins and the induction of inflammatory and immune responses. Highly specialised M cells, responsible for the immune sensing of luminal bacteria, take up microorganisms and macromolecules and deliver them to lymphoid tissues for efficient mucosal immune responses [23]. Under normal circumstances, the symbiotic flora and the intestinal immunity of the host work together for the homeostasis of the intestinal barrier. Through PRR signalling, intestinal commensal bacteria prime the immune system by fostering cytokine production by epithelial cells to promote the maturation of dendritic cells (DCs) and macrophages with tolerogenic properties [24,25]. In addition, resident host microflora condition mucus production by caliciform cells [26] or by the expression of AMPs. Under homeostatic conditions, IECs do not respond to Toll-like receptor (TLR) stimuli, whereas increased TLR expression has been observed in the inflammatory response. In fact, TLR signalling in the gut is involved in both tolerance induction and promoting intestinal inflammation [27]; fosters epithelial cell proliferation, the generation of IgA and the expression of antimicrobial molecules [28] and helps the gut immune system to generate homeostasis in the intestinal barrier [29]. The activation of TLRs affects the expression of tight junction proteins [30]. Inflammatory processes in the intestine induce the disruption of epithelial junctions, which promotes microbial translocation and the release of their toxic metabolites into the lamina propria, heightening the inflammatory injury and inducing enterocytes’ dysfunction. The resulting alteration in gut permeability may account for malabsorption, lack of micronutrients and a compromised excretion of waste products [31].

### 2.2. The Role of Microbiota and GALT

The microbiota in the GI tract regulates the local systemic immune response, influencing the development and reactivity of the GALT from an early age [32]. Early microbial colonisation determines the development of humoral immune mechanisms related to the local production of secretory IgA and IgM, which are particularly important in the early phase of the immune response and for the development of food tolerance [33,34]. The gastrointestinal (GI) tract contains a large population of T and B cells, harbouring over 90% of all lymphocytes [33,34,35].

The balance of different subpopulations of T-helper (Th) lymphocytes is crucial. After antigen recognition (“priming”), activated B and Th lymphocytes migrate and actively proliferate, with local cytokine and IgA production. Antigen priming results in the “cascading” production of different cytokines by the activated Th cells, which can result in two types of immune responses, predominantly proinflammatory, and therefore potentially harmful, or anti-inflammatory [33,34,35,36,37]. The GALT has to fulfil seemingly conflicting functions: (i) it must be tolerant of the overlying luminal microbiota, (ii) prevent the induction of an exuberant immune response, (iii) control its overgrowth and (iv) prevent bacterial translocation from the lumen to the mucosa with possible systemic dissemination. The central role of the microbiota in the development of GALT-mediated immune defences is not unexpected, considering that the intestinal mucosa constitutes the largest contact surface with antigens from the lumen and that the resident microflora lining the mucosa of the gut normally accounts for most of the antigens presented to resident immune cells.

Besides lymphocytes, myeloid cells—in particular, DCs and macrophages—also maintain gut homeostasis and are essential to mediate immune tolerances to food antigens and commensals. In a healthy gut, DCs migrate from the lamina propria to drain mesenteric lymph nodes (MLNs) or T-cell zones of Peyer’s patches and determine the outcome of antigen-specific adaptive immune responses by priming naïve T cells and induce tolerogenic intestinal T-cell responses [38,39,40]. Notably, CD103+ DCs have important homeostatic functions, due to their ability to induce intestinal Tregs and noninflammatory mucosal Th17 cells [40]. In addition to DCs, intestinal macrophages can express MHC class II and may participate in T-cell priming, although they are highly specialised to avoid overt immunity in response to the gut microbiota [39]. Mucosal macrophages are also involved in the maintenance of the epithelial barrier integrity, as well as in the protection from excessive inflammation, by efficient phagocytosis and the killing of organisms that breach the epithelium [41]. 

HIV infection may result in the disruption of the GI tract by affecting the microbiota and by inducing substantial damage to the epithelial barriers and GALT with the massive depletion of CD4+ T cells, mucosal inflammation and aberrant cytokine production [17,18,38]. During HIV infection, gut mucosal macrophages and DCs exhibit reduced phagocytic activity and defective clearance of microbial products that cross the damaged intestinal barrier, contributing to local inflammation and to an impaired mucosal barrier function [42,43,44,45]. Moreover, gut DCs may drive mucosal immune activation and inflammation during HIV-1 infection. It has been hypothesised that intestinal DCs may have a critical role in mediating viral and bacterial signals during HIV-1 infection in vivo. Indeed, Dillon et al. found that the altered colonic mucosal microbiota related to HIV infection—in particular, an increase in *Prevotella* species—induced a dysregulation of the intestinal myeloid DC activation phenotype and production of proinflammatory cytokines, leading to additional nonspecific T-cell activation and to the expansion of Th1, Th17 and Th22, targets for viral replication [46]. The subsequent infection and depletion of these cells may contribute to epithelial barrier disruption and microbial translocation, thereby potentiating a vicious cycle that ultimately leads to systemic inflammation and immune activation. In another model, microbial products such as LPS stimulate tissue-resident dendritic cells to produce interferon-alpha and activate the IDO pathway [47]. IDO plays a critical role in regulating the Treg/Th17 balance, inducing Foxp3 expression and Treg development and suppressing the Th17 lineage [48,49]. This loss of Th17 cells leads to even more microbial translocation and contributes to the persistent inflammatory state observed in tissues during chronic HIV-1 disease.

Impairment of the intestinal integrity promotes the translocation of bacterial products such as lipopolysaccharides into the bloodstream, causing the systemic activation of T lymphocytes and higher viral replication and the risk of comorbidities and facilitating disease progression in PLWH [17,18,42]. The first histological studies identified a preferential depletion of memory CD4+ T cells in the gastrointestinal biopsies of PLWH [50,51]. Because of the difficulty in carrying out studies on tissue compartments in people who recently acquired HIV, reasonable knowledge on CD4+ T-cell dynamics and the contribution of gut tissue to HIV replication and persistence have been obtained from an animal model. Upon the mucosal challenge of *macaques* infected with simian immunodeficiency virus (SIV), high levels of virus replication and CD4+ T-cell depletion in the GALT were observed that took place already at the earliest stages of infection, with evidence of seeding even when ART was initiated three days after viral infection [52]. The findings from the animal model, while informative, cannot be directly extrapolated to humans, as the characteristics of the virus and host significantly differ; thus, little is known about the mechanism of the formation of the viral reservoir in humans.

Studies in PLWH on ART have reported that HIV DNA levels in CD4+ T cells were 2- to 12-fold higher in the duodenum, ileum, right colon and rectum as compared to the peripheral blood, probably due to a higher percentage of CD4+ T cells expressing the HIV coreceptor CCR5, upregulated by lipopolysaccharides of the gut microbiome, with an increased susceptibility to infection [53]. Based on these findings, it is possible to speculate that the gut harbours 80–95% of all HIV-infected cells in the human body, and thus, it provides an important reservoir for disease persistence with important implications for clinical therapy.

While HIV DNA latency in CD4+ T cells in the GI tract is well documented, the contribution of mucosal myeloid cells in latent HIV infection and spreading of the virus is still under debate, as they account for only 4% of the total HIV DNA [54]. HIV can persist in tissue macrophages, even with their low permissiveness to HIV infection after ART treatment [54]. Gut mucosal myeloid cells represent a pivotal site of potential HIV carriers due to their role as a portal of entry and of transmission to lymphoid cells. Indeed, during mucosal viral exposure, macrophages and DCs selectively engulf HIV due to higher expression of CCR5 and DC-SIGN, respectively, and are able to transfer the virus to T cells [55].

Different subsets of CD4+ T cells in the intestinal mucosa, i.e., Th1, Th2, Th17, Th22, regulatory T cells (Treg) and follicular T-helper cells (Tfh), play a role in the defence against intestinal microbial pathogens and in maintaining the integrity of the mucosal barrier and homeostasis [56,57,58,59,60,61,62]. Differently from the peripheral blood where naïve and central memory (Tcm) constitute the largest populations, the gut contains primarily the effector memory (Tem) and transitional memory (Ttm) T-cell subsets [56,57,59]. The role of the main T-lymphocyte subsets in the establishment of the HIV viral reservoir in the gut is briefly discussed below.

### 2.3. T-Cell Subsets as Viral Reservoir in the Gut Mucosa

The GI tract harbours a diverse subset of CD4+ T cells with distinct phenotypic characteristics, functions and memory differentiation status. Notably, several studies have reported that, in both untreated and ART-treated patients, the HIV DNA and RNA levels in CD4+ T cells are higher in the GI tract as compared to the peripheral blood [61,62,63]. Therefore, characterising the cell types that support HIV persistence and the role of the GI environment is critical to understanding how latency may be established and to identifying cellular targets for HIV eradication strategies.

T-helper 17 (Th17) cells represent 80 to 90% of the total CD4+ T cells and are well defined by their ability to secrete cytokines such as IL-17, IL-21, IL-22 and IL-26. Th17 cells are a subset defined by the expression of C-C motif chemokine receptor 6 (CCR6), which plays a role in cell migration into different tissues, including the intestinal mucosa, CCR4 (CD194), C-type lectin-like receptor CD161, signal transducer and activator of transcription 3 transcription 3 (STAT3) and transcription factor RORC2, which is an isoform of ROR-γ (nuclear receptor ROR-gamma) [64] (Figure 1).

Th17 cells play, in homeostatic conditions, a pivotal role in the mucosal integrity, inducing the proliferation of enterocytes and, in defence, producing defensins and recruiting neutrophils in response to intestinal infections [65]. The modulation of inflammation occurs through the production of IL-17 and IL-22 and by attracting inflammatory cells and creating a favourable environment for viral replication [64]. Th17 are the main targets for viral reservoir formation in the GALT, showing greater permissiveness to the virus and a higher level of integrated HIV DNA than Th1 and Th2 in vivo [65,66,67]. Indeed, it has been reported that the HIV DNA levels in the sigmoid, which remain high despite long-term suppressive therapy, correlate negatively with the sigmoid Th17 frequency [68]. Similar findings were reported in another study performed on CD4+ T cells expressing Th17 and CCR6 cell markers and isolated from sigmoid colon biopsies of ART-treated PLWH, which carried integrated HIV-1 DNA [66,69,70]. Thus, CCR6 is a marker for the majority of HIV-infected CD4+ T cells in colon tissue [66]. Th17 cells have also been described as long-lasting cells with increased abilities to self-renew, and for that reason, they may serve as a preferential niche for HIV-1 long-term persistence [65,71].

Following HIV-1 infection, Th17 cells are rapidly depleted at blood and mucosal sites, causing an imbalance of the Th17/Treg ratio, microbial translocation and systemic inflammation and maintaining the HIV reservoir in PLWH on ART and, most importantly, hampering viral eradication [68,72,73]. ART has been shown to improve Th17 reconstitution in blood; conversely, in the intestinal mucosa, the recovery of Th17 function is highly variable and depends on ART starting; the earlier it starts, the earlier there will be Th17 cell recovery [69,70]. 

Of note, a subset of CD4+ T cells producing IL-17 and IFN-gamma with a CCR6+CCR4-CXCR3+ phenotype are termed Th1/Th17 [74] and exhibit high levels of cellular activation, contributing to their increased permissiveness to HIV infection [62]. Moreover, a previous work indicated that Th1/Th17 harbour high levels of HIV-1 DNA in ART-treated PLWH and that the levels of CCR6+ memory T cells are higher in the colon than in peripheral blood [62,65]. In addition, other subsets of Th17 cells, i.e., CCR6+ CXCR3- CCR4- and CCR6+CXCR3+ CCR4+ cells, have been indicated as HIV reservoirs, because they contain HIV DNA and are able to produce the infectious virus [74,75].

T-helper 22 (Th22) cells express CCR10, CCR6 and CCR4 chemokine receptors and, by producing IL-22, support intestinal epithelial barrier integrity and function, promoting the release of antibacterial defensins and mucins [76] and stimulating epithelial cell renewal. In the gut of healthy individuals, the levels of Th22 cells are lower than Th17 cells [77]. Following HIV infection, Th22 cells in the intestinal mucosa are, to a great extent, depleted, and their frequency appears to be rescued with suppressive ART in the lamina propria of the duodenal mucosa of PLWH [78]. However, the contribution of Th22 cells to the HIV reservoir in the GI tract has not been fully investigated. Nayrac et al. reported that Th22 cells in the gut express lower levels of CCR5 compared to Th17 cells and that, in gut histocultures ex vivo, Th22 cells are less susceptible to the R5-tropic virus than Th17 cells, showing lower levels of infection [78]. 

Regulatory T cells (Tregs), expressing the transcription factor FoxP3, exert immunosuppressive activity against microbes and may also control inappropriate or exaggerated immune activation, causing a decrease in the antiviral immune response and facilitating disease persistence and progression. Furthermore, during a viral infection, Tregs inhibit virus-specific T-cell responses [79]. It is as if they are two sides of the same coin, as Tregs may prevent immune hyperactivation caused by HIV infection, but they can also suppress specific immune responses against HIV [80]. During HIV infection, the consequences of Treg expansion can be either beneficial, by preventing generalised T-cell activation, or detrimental, by suppressing HIV-specific responses and thus contributing to viral persistence. Moreover, HIV-1 can directly infect Tregs, altering their phenotype by the downregulation of Foxp3 and CD25, and function, impairing their suppressive capacity [80,81]. The characteristic of Treg cells are their longevity and resistance to apoptosis; thus, following HIV infection, they could form latent reservoirs of the virus and significantly hinder HIV-1 eradication. Several authors have supported the existence of latently infected Tregs in peripheral blood; however, in the absence of studies on Treg HIV latency in human tissue compartments, a SIV monkey model provided evidence of Tregs as a reservoir of the virus. They described an increase in the number of infected Tregs in the GALT in a chronic untreated infection that was suggestive of a latent Treg reservoir in lymphoid tissue, although more direct evidence is required to support Treg as an important HIV reservoir in the gut [82,83].

Follicular T-helper cells (Tfh) are a specialised CD4+ T-cell subset that express high levels of CXCR5, a follicular homing receptor, and transcription factor B-cell lymphoma 6 (BCL6), as well as CXCR3, PD1 and ICOS [84]. Tfh cells are involved in the induction and differentiation of B cells into plasma cells and memory cells and specifically reside in the germinal centres of secondary lymphoid organs, where they have been imputed to be a persistent reservoir of latent and productive HIV [62]. Tfh cells are highly infected by HIV in vitro and in PLWH during suppressive ART, in which lymph nodes, as well as PD-1 Tfh cells, can produce a replication-competent virus, sustaining the contribution of GI Tfh cells as a potential HIV reservoir in vivo. Indeed, Tfh cells in lymph nodes exhibit high levels of HIV DNA [62,85,86]; a monkey model study showed that Tfh cells isolated from the jejunum of *Rhesus macaques* harboured higher levels of SIV DNA than Tcm [87]. Further evidence from humanised mouse models of HIV supported the contribution of GI Tfh cells as a cellular HIV reservoir [88]. Together, these data underscore the importance of Tfh as a potential HIV reservoir in the GI tract. The intense exposure of Tfh cells to the virus on follicular dendritic cells (FDC) and low frequencies of virus-specific cytotoxic T lymphocytes (CTL) in B-cell follicles lead to a high level of infection of Tfh cells [84]. The expansion of HIV-specific Tfh cells can lead to compromised B-cell responses such as hypergammaglobulinemia, as Tfh cells are needed to promote class switching and the affinity maturation of antibody-producing B cells [85]. During HIV infection, dysfunctional Tfh cells in the germinal centre (GC) of Peyer’s patches in the GI tract alter B-cell responses, especially the secretion of IgAs that drives alterations in the intestinal microbiota and immune homeostasis and, as a consequence, in microbial translocation and immune activation [85,89]. 

Gamma delta (γδ) T cells present a T-cell receptor (TCR) composed of a γ and a δ chain and reside both in the peripheral blood and in mucosal tissues [90]. The Vδ2 subset is the main subtype (90%) of γδ T cells in the peripheral blood, and the Vδ1 and Vδ3 subsets are present in the tissues and mucosa, respectively [90]. The γδ cells exert protective functions against pathogens, including HIV, although they may be dysregulated during HIV infection [62,91]. Gamma delta T cells are mainly tissue-resident and very abundant populations in the gut. Localised between the intestinal epithelial cells, γδ cells preserve the mucosal integrity and maintain symbiosis with the microbiota [90]. Otherwise, γδ T-cell subsets can reside in the lamina propria and circulate in the blood or be tissue-resident, but differently from the intestinal intraepithelial γδ, they produce IL-17, depending on the local microbiota. Finally, γδ T cells can be present in Peyer’s patch inside the GC. In mucosal tissues of chronically PLWH, Vδ1 T cells are severely diminished [62,92]. γδ T cells in the peripheral blood of ART-treated patients harbour replication-competent HIV-1 DNA, and even following the reconstitution of CD4+ T cells, the frequency of these cells does not return to the pre-infection state [93]. To date, the contribution of γδ T cells as a HIV reservoir is still unknown. 

Memory T cells can be broadly divided into phenotypically distinct central (Tcm; CD45RAneg CCR7+ CD62L+), transitional (Ttm; CD45RAneg CCR7neg CD95+ CD62L+), effector (Tem; CD45RAneg CCR7neg CD95+ CD28neg CD62Lneg) and tissue-resident memory T cells (Trm; CD45RAneg CCR7neg CD69+ ± CD103+), depending on their expression of surface markers, location and function [94,95,96]. Tcm cells circulate through lymphoid tissue and the blood and are capable of regeneration and long-term maintenance, whereas Tem cells are more present in peripheral tissues and have effector functions at sites of inflammation. In addition, transitional memory T (Ttm) cells exhibit functional characteristics that are intermediate to Tcm and Tem [96]. Trm cells permanently reside in tissue and do not recirculate via the bloodstream. Most memory T cells in the gut are primarily Trm with distinct phenotypic and functional properties compared to their blood counterparts [59,96]. Trm cells in mucosal and peripheral tissue express CD69, which contributes to T-cell retention in tissues and is expressed by more than 80% of the CD4+ T cells in the colon [59].

In contrast to the peripheral blood, where Tcm and Ttm cells have been identified as major contributors to the latent HIV-1 reservoir [97], in the GI tract, especially in ileum and rectum, Tem cells harbour the highest level of HIV DNA and RNA [98]. Conversely, the proportion of Tcm cells in the upper intestine is very low; thus, they do not contribute significantly to the HIV reservoir. Furthermore, highly activated CD4+ T cells for exposure to antigens from the GI tract, such as Tem, Trm and Ttm, provide a pool of ongoing targets for productive HIV infection [53,98]. Interestingly, it has been described that Trm cells respond to tissue-specific inflammatory factors, which can induce, through the production of transforming growth factor β (TGF-β), high expression of the CD103 integrin on this subset of cells [99]. Although, in the GI tract, a small number of Trm cells are CD103+, this subset expresses CCR5, suggesting that they may contribute to HIV latency [100]. It has been described that, in PLWH on ART, the HIV DNA levels in CD4+ T cells in both the ileum and rectum are higher as compared to the peripheral blood [53], suggesting that Ttm may also play an important role in HIV persistence in the gut. Also, the stem cell-like memory T-cell subset (Tscm), a rare subset of memory lymphocytes endowed with the multipotent capacity to reconstitute memory and effector subsets, may contribute to HIV-1 persistence, and even if Tscm cells constitute a tiny part of the total reservoir in PLWH on ART, HIV-1 DNA is present in Tscm at high levels in the peripheral blood [101,102]. The frequency of Tscm cells in the gut is low, and their role in maintaining the gut HIV reservoir still needs to be clarified. Although further research is necessary to assess the contribution of the CD4+ T-cell subsets as the cellular HIV reservoir in the GI tract, all these observations suggest that the mechanisms required for maintaining the HIV reservoir in the GALT are different from those required in the blood.

Understanding the mechanisms explaining a direct correlation between HIV DNA levels in the GALT, microbial translocation and systemic inflammatory markers could inform about the pathogenesis and treatment of dysbiosis in HIV infection. Different studies have explored the link between the HIV gut reservoir and microbial translocation. d’Ettorre et al. showed that the LPS levels correlate with the HIV DNA levels in rectosigmoid biopsies of chronically treated PLWH [50]. This finding could be explained, since HIV infection and replication at the intestinal mucosa leads to immune activation and tissue inflammation with a disruption of the intestinal barrier’s integrity, resulting in higher markers of bacterial translocation in plasma.

## 3. Effects of HIV Infection on Gut Microbiota and Systemic Immune Activation

### 3.1. The Healthy Microbiota

The human gut microbiota is involved in a wide range of physiological, metabolic and immunological functions, including the maintenance of the gut epithelial barrier integrity and the interactions with different immune cells in the gut, promoting the development of mucosal immunity [103,104]. A healthy microbiota comprises primarily four groups of bacteria, including Actinobacteria, Firmicutes, Proteobacteria and Bacteroides, with Bacteroides and Firmicutes dominating more than 90% of the total bacteria inhabiting the human gut [105]. The gut microbiota may influence the innate and adaptive immunity. The evidence suggests that Bacteroides spp. are capable of producing anti-inflammatory cytokines by reducing the immune activation [106], of stimulating intestinal dendritic cells and plasma cells in the intestinal mucosa to secrete IgA and are inducers of regulatory T-cell functions [107]. A well-balanced gut microbiota also maintains the intestinal epithelial barrier integrity through the restoration of tight junction proteins, formation of antimicrobial proteins and inhibition of epithelial cells binding with pathogenic bacteria. Thus, immune responses mediated by the gut microbiota are essential for preventing the intestinal permeability [108]. Notably, the function, development and differentiation of mucosal T cells residing within the lamina propria are affected by commensal microbial signals and also by short-chain fatty acids (SCFAs) and metabolites of the gut microbiota [108].

### 3.2. The Effects of HIV-1 Infection on Gut Microbiota

HIV-1 infection within the gut is associated with enteropathy, aberrant cytokine production, mucosal inflammation and damage to intestinal epithelial cells. HIV leads to the disruption of tight junctions and cellular death and to impairment of the intestinal epithelial barrier [16,17]. This picture is associated with inflammation, as it allows the movement of bacterial products and perhaps even viable bacteria through the lamina propria and into the systemic circulation [16,17]. In PLWH even on effective ART, the composition of the gut microbiota differs from that of healthy people. PLWH present an increase in potentially pathogenic bacteria that promotes chronic inflammation and immune dysfunction and a decrease in commensal beneficial bacteria [42,109,110]. At the phyla level, PLWH have reduced Firmicutes and Bacteroides and are enriched with Proteobacteria, even under ART [103]. Proteobacteria include some pathogenic bacteria (e.g., Salmonella, Acinetobacter, Escherichia and Yersinia), while Firmicutes include many groups of Gram-positive species (Staphylococci, Streptococci and Lactobacilli) with anti-inflammatory and immune-regulatory properties. Specifically, it has been observed that the high relative abundance of the Prevotella strains in PLWH may result in an altered metabolism of tryptophan with activation of the kynurenine/indoleamine 2,3-dioxygenase (IDO) pathway and may enhance Th17-mediated mucosal inflammation and the loss of Th17 cells, promoting a proinflammatory status and compromising the integrity of the GI tract [17,106,111,112]. An altered tryptophan metabolism has been correlated with immune activation and increased mortality in PLWH [113]. Also, species of Prevotella are involved in proinflammatory pathways mediated by lipopolysaccharide (LPS) synthesis. LPS binds CD14 on the surface of monocytes and macrophages, promoting the activation of Toll-like receptor 4 (TLR4) and the production of proinflammatory cytokines [28,31,106]. Interestingly, Enterobacteriaceae, members of Proteobacteria, are able to utilise, as energy sources in their respiratory chains, reactive oxygen species (ROS) produced by neutrophils and macrophages during inflammation in both acute and chronically PLWH. This confers a competitive growth advantage over endogenous gut microbiota, exacerbating gut inflammation and allowing translocation across the gut barrier [114].

Conversely, Bacteroides, which are reduced in PLWH, correlate with the production of anti-inflammatory cytokines in the GI tract, leading to a protective effect against inflammation [106,115]. Also, reduced Firmicutes, found in PLWH, contribute to decreased levels of anti-inflammatory short-chain fatty acids (SCFAs) such as butyrate, which is important in regulating the intestinal homeostasis and barrier integrity [106]. 

The change in the GI tract microbial diversity also influences the immune status in the gastrointestinal tract of PLWH. Following the depletion of Th17 cells and the significant increase in Tregs in PLWH, a dysregulation of the Treg/Th17 balance occurs, highlighting the important role of the microbiota in the induction and regulation of Tregs [116]. These alterations induce the loss of tight junctions between enterocytes and the translocation of microbial products to the systemic circulation, inducing chronic immune activation and the suppression of viral clearance [117]. Moreover, patients with lower CD4+ T-cell counts present less richness and variety of the gut microbiota [118], and the activation of intestinal mucosal T cells and DCs was positively correlated with an abundance of Prevotella strains, causing an increased mucosal viral load and systemic immune activation [119].

In conclusion, dysbiosis and microbial translocation in PLWH are associated with systemic immune activation [120], and this predicts disease progression in untreated and ART-treated PLWH. This chronic inflammatory environment, due to the compromised structural integrity of the mucosal barrier, appears to cause fibrosis in lymphoid tissues, creating an unfavourable microenvironment for the survival of naïve T cells, which, in turn, causes CD4+ T-cell regenerative failure or the delayed reconstitution of CD4+ T cells and HIV disease progression [121,122]. In PLWH, the loss of germinal centre architecture, follicular damage and T-/B-cell apoptosis in Peyer’s patches occur early and persist through chronic viral infection. Estes et al. found that the degree of CD4+ T-cell restoration in Peyer’s patches was better when antiretroviral therapy was started earlier rather than later in the course of infection during which immune activation and inflammation contribute to collagen deposition and disruption of the lymphatic tissue architecture [123]. Another study showed that, early in HIV infection, the structure of Peyer’s patches was specifically modified, lymphoid follicles showed signs of damage and T- and B-cell apoptosis and, most importantly, the induction of polyclonal B-cell differentiation could explain the reduction in both HIV-1-induced antibody responses [124]. Systemic fibrosis, through mechanisms including oxidative stress and mitochondrial dysfunction, may also cause accelerated senescence in PLWH [125]. Notably, chronic immune activation drives an increased turnover of CD4+ T cells due to higher T-cell activation and dysregulated cytokine production, with the generation of CD4+ Tem cells from Tcm cells, thereby maintaining HIV infection at the peripheral and tissue levels [126].

If microbial translocation persists during effective ART, it also predicts mortality and is associated with several comorbidities such as cardiovascular and metabolic diseases and neurocognitive dysfunction.

## 4. Role of HIV-Related Gut Dysbiosis in Non-AIDS Comorbidities

The enteric microbiota alterations, dysbiosis, observed in HIV-infected people have far-reaching consequences on other organs, such as the liver, brain, adipose tissue, muscles and pancreas, altering their homeostasis and contributing to organ injury in metabolic diseases [127]. The most common risk factors underlying non-AIDS comorbidities in PLWH are chronic immune activation and inflammation [120]. Intestinal barrier disruption, which occurs after rapid loss of the gastrointestinal mucosal integrity due to HIV replication, allows microbial components to enter from the gut lumen into the circulatory system, inducing persistent immune activation [128]. Indeed, HIV disease progression is also strongly associated with changes in the gut microbiota, creating a vicious cycle of inflammation and immune activation between the viral and bacterial factors (Figure 2) [129]. 

### 4.1. The Gut–Liver Axis and Liver Disease

In the context of non-AIDS-related comorbidities, the gut–liver axis has emerged as a potential pathway contributing to liver disease in PLWH. Microbes and their metabolites, after crossing the epithelial barrier, travel to the liver, passing through the portal circulation, and upregulate the inflammatory and fibrotic pathways [125,130]. Hepatic damage occurs primarily via increased hepatic levels of bacterial LPS that binds to TRL-4 on Kupffer and hepatic stellate cells (HSCs), increasing the transcriptional activation of proinflammatory genes and cytokines, such as IL-1β and TNF-α, in the liver via the MyD88 and NF-κB pathways and interferon (IFN) regulatory factors [125,131]. LPS may also induce hepatic injury by promoting hepatocyte cell death [125,132]. Additionally, systemic inflammation, due to gut microbial translocation during HIV infection, may cause liver fibrosis through mechanisms including oxidative stress, mitochondrial dysfunction, increased lipid peroxidation and the reduced expression of IFN-gamma, which is an antifibrotic cytokine inducing the apoptosis of activated HSCs [125,133].

### 4.2. The Cardiovascular and Metabolic Diseases

Other pathologies that may be related to the dysbiosis and chronic immune activation in HIV infection include cardiovascular diseases (CVDs). The persistent inflammation in PLWH under ART is correlated with endothelial dysfunction and atherosclerosis, leading to a propensity to develop CVDs, as well as cerebral and peripheral vascular diseases [134,135,136]. Concerning the link between gut dysbiosis and CVDs in PLWH, it has been found that trimethylamine-N-oxide (TMAO), which is formed in the liver from TMA, exerts proatherogenic activity, and its production is mostly mediated by bacteria enriched during HIV infection, such as Proteobacteria [103,137], and is associated with the presence of silent ischemia, coronary plaques and acute myocardial infarction in PLWH [136,137,138,139,140]. TMAO has been reported to affect cholesterol and bile acid metabolism, promoting foam cell formation [141] and activating monocytes with a consequent increase of inflammatory biomarkers such as LPS and sCD14 among PLWH and of the risk of atherosclerosis [103,142]. Indeed, it has been reported that the sCD14 and IL-6 levels in the sera of PLWH correlated with the common carotid intima media thickness [134,143]. Similarly, other studies showed that the serum levels of LPS and sCD14 were correlated with subclinical atherosclerosis progression and with coronary artery calcification in PLWH [144,145]. GI microbiome-derived TMAO can also activate the thioredoxin (TRX)-interacting protein (TXNIP)-NLRP3 inflammasome pathway, which is correlated with insulin resistance and CVDs [135,146]. Moreover, an altered tryptophan catabolism through the kynurenine pathway, induced by a Proteobacteria-enriched and Bacteroidia-depleted microbiota, could be predictive of increased CVDs in PLWH [113,134]. In fact, an elevated kynurenine-to-tryptophan ratio was observed in PLWH with atherosclerosis [147,148]—in particular, in those developing carotid artery atherosclerosis [149,150]. A higher plasma KYN/TRP ratio also correlates with endothelial dysfunction and inflammation in PLWH with type 2 diabetes (T2D), further supporting a link between a high KYN/TRP ratio and metabolic disorders in PLWH [103]. Interestingly, the disturbed gut microbiota composition found in PLWH is similar to that observed in subjects with T2D and/or obesity, especially for the reduction of butyrate-producing bacteria involved in anti-inflammatory responses, in the regulation of intestinal homeostasis and of the metabolism of fatty acids with alterations that may induce oxidative stress, promoting cardiometabolic-adverse manifestations [103,151,152]. In fact, in the context of HIV infections, recent studies have reported an inverse relationship between the KYN/TRP ratio and butyrate producers and an association with metabolic disorders such as diabetes, hypertension and dyslipidaemia [103,153,154]. On the other hand, the positive association with bacteria that flare up inflammation by inducing proinflammatory cytokines and LPS was linked to insulin resistance (IR) and lipid alterations in PLWH, such as HIV-associated lipodystrophy and ‘wasting syndrome’, metabolic disorders characterised by the impaired distribution of body fat and by malabsorption and low weight gain, respectively [106,134,155,156]. Insulin resistance and LPS, by altering the vascular homeostasis, were also associated with hypertension in PLWH [157]. Moreover, it has been shown that LPS and sCD14 are negatively correlated with high-density lipoprotein (HDL) cholesterol, contributing also to the increased risk of CVDs in PLWH [131,158]. Collectively, these findings suggest that inflammation and oxidative stress exerted by alterations in the gut microbiota are directly associated with alterations in cardiovascular functionality in PLWH and may indirectly affect the major risk factors for CVD, including obesity, type 2 diabetes and metabolic syndrome [134,140]. Systemic lipid dysregulation is also associated with immune cell dysregulation—in particular, with CD4 T-cell activation and altered HIV-specific T-cell responses as a glycolytic phenotype is induced that increases HIV infectivity and HIV reservoir seeding. The reduced functionality of the T-cell compartment is also ascribed to T-cell exhaustion [159]. Indeed, HIV-related dysbiosis may modulate HIV latency, contributing to the establishment of viral sanctuaries resistant to ART [134]. Moreover, the aforementioned Treg/Th17 imbalance in PLWH contributes to an immunosuppressive microenvironment, thus favouring HIV escape immune surveillance [134].

### 4.3. The Brain–Gut Axis and Neurocognitive Disorders

Last, but not least, a series of largely preclinical and clinical observations have shown a constant interaction within the microbiome–gut–brain axis. The gut microbiota and the central nervous system communicate through immune, endocrine and nervous signalling mechanisms [160]. The disruption of the normal gut microbiota composition during HIV infection may interfere with brain–gut–microbiome processes, contributing to cognitive impairment and neurocognitive disorders in PLWH [161]. The mechanisms proposed to explain this theory include inflammation of the human central nervous system (CNS) through stimulation of the innate immunity by LPS and CD14 increased due to microbial translocation and white matter abnormalities with consequent neurocognitive dysfunction in this population [162,163]. Also, decreased SCFA production, which affects the increased intestinal permeability, may impact the permeability of the blood–brain barrier in a similar way [164]. In fact, lower levels of Firmicutes in the gut have been associated with HIV-related dementia (HAD) [165]. Other hypothesised mechanisms are molecular mimicry to bacterial antigens leading to dysfunctional adaptive immune responses, the production of bacterial metabolites that may directly damage the brain and were supposed to be related to specific neurodegenerative disorders, including Alzheimer’s disease, and vagal dysfunction related to HIV-associated autonomic neuropathy, associated with intestinal bacterial overgrowth and changes in immune and gastrointestinal functions [161,166,167,168]. Moreover, the vagus nerve is essential for reducing intestinal permeability in HIV infection, thus preventing microbial products from penetrating the lamina propria and entering the circulation to amplify the immune dysregulation and inflammation [169]. HIV neuropathogenesis may also occur through the mechanisms of oxidative stress and metabolic disturbances indirectly linked to gut bacterial processes [161], and neuroinflammation is exacerbated by the persistent replication of HIV in CNS unhindered by ART, as the penetrability of these compounds through the blood–brain barrier is variable and incomplete [170]. Therefore, nearly half of all PLWH may be affected by HIV-associated neurocognitive decline (HAND) [161].

## 5. Conclusions

In recent years, there has been a great effort in characterising how the significant compositional gut microbiota modifications in PLWH may be involved in the inflammatory processes during HIV infection and their influence on HIV metabolic disorders. Moreover, understanding the contribution of the GALT to the HIV latent reservoir and the relationship between HIV persistence and the gut microbiome will be critical to developing new treatments, such as probiotics or postbiotics, to restoring the GI tract health, eliminating HIV reservoirs and to preventing and managing HIV-related comorbidities. In this review, we describe that high-level viral replication in the GALT of PLWH and the loss of CD4+ T cells result in the disruption of the mucosal barrier, microbial translocation, persistent inflammation and local and systemic immune activation, contributing to organ damage. Even in those ART-treated, PLWH still persists in the GALT, and eradication is still out of reach due to latent viral reservoirs in the gut-associated lymphoid tissues. Further studies are warranted regarding the frequency, phenotype and functional status of CD4+ T-cell populations in the intestinal mucosa to provide new insights into the role of the GI tract in HIV persistence, which is pivotal for new drug designs, for reducing the HIV reservoir, immune activation and disease progression.

Moreover, it would be interesting to better evaluate the impact of different ART regimens on the microbiota composition and their relationship with the GALT and HIV reservoir dynamics. Indeed, long-acting ART regimens could be a promising solution in terms of reducing the gut dysbiosis by shunting the gut passage and therefore decreasing microbial translocation and systemic immune activation. This may theoretically act on residual HIV viral replication and on HIV reservoirs and give new insights into the therapeutic efficacy on gut inflammation during HIV infection. In addition, improving knowledge about microbiota species and metabolites associated with HIV infection and/or progression will be crucial to evaluate therapeutic strategies targeting the microbiota. This may be helpful in the management of chronic HIV infection and in the prevention of comorbidities that are associated with immune activation and microbial translocation. In the future, a larger use of probiotics as adjuvant treatments or dietary components for PLWH and also as personalised treatment approaches will represent a beneficial strategy to manage gut dysbiosis, enhancing gut mucosal immune responses and facilitating HIV clearance.

## Figures and Tables

**Figure 1 ijms-24-12193-f001:**
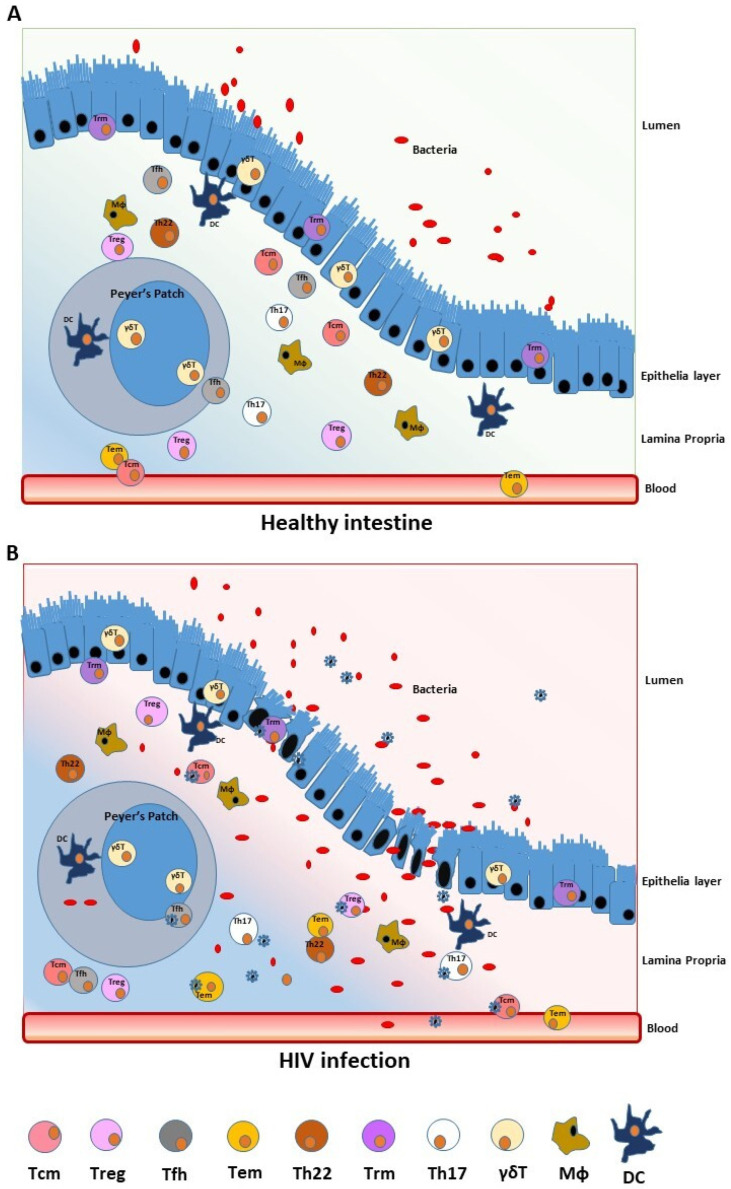
Effect of HIV infection on healthy intestines. The gut mucosa is composed of epithelial cells; tight junctions; mucus layers and a highly regulated complex of T and B lymphocytes, macrophages and dendritic cells. The gut microbiota regulates the local immune response, and systemic exposure to microbes and microbial metabolites is limited (**A**). HIV infection leads to a loss in tight junctions, alteration of the gut microbiota and microbial translocation, depletion or dysregulation of resident immune cells and establishment of a HIV viral reservoir in CD4+ T-lymphocyte subsets (**B**).

**Figure 2 ijms-24-12193-f002:**
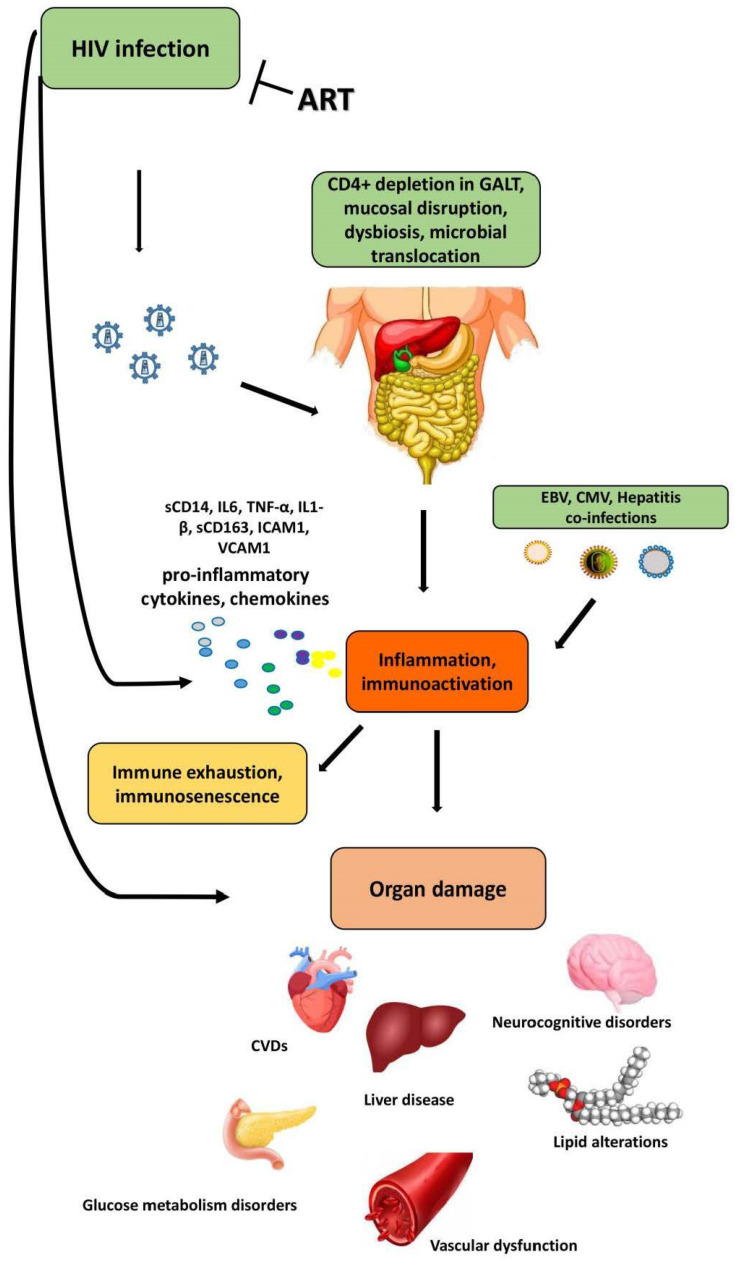
Systemic consequences of HIV infection in the GALT. HIV infection causes, even in ART-treated PLWH, massive CD4+ T-cell depletion, gut barrier dysfunction and microbial translocation. The collective effect of these insults is systemic inflammation and chronic immunoactivation, increased by the production of proinflammatory cytokines and chemokines, which, in turn, result in immune exhaustion and immunosenescence. These processes lead to organ damage and associated comorbidities.

## Data Availability

Not applicable.

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
