# Peer review of "HIV-1–Host Interaction in Gut-Associated Lymphoid Tissue (GALT): Effects on Local Environment and Comorbidities"

_ijms, 2023, doi:10.3390/ijms241512193_

Round 1

Reviewer 1 Report

In this review by Moretti et al. the authors explored several aspects of HIV pathogenesis and the gut tissue, from cell reservoir and its interaction with the microbiome to HIV-associated comorbidities.

The review is well-structured and written. It gives the reader a broad understanding of a topic and provides up-to-date references if they want to explore particular topics. 

One critical comment is about some terminology. Please avoid (and I suggest changing it) using "HIV-positives individuals", "HIV-infected people", "HIV-infected patients" and instead use PLWH.

I would also suggest changing a bit the colors in Figure 1. The green and the red are too dark and make it hard to see the cell label. Also, the cell name is very small, and it is a bit hard to differentiate Tcm from Tem. Maybe you can try to add a legend beside the figure rectangle for each cell type to make it "cleaner" and easier to look at. 

A minor comment, check carefully for typos (example in line 33). Also, many words have a dash in the middle, for example in lines 35, 143, 183, and so on. It could have happened in the submission and formatting process, but it needs to be carefully revised since there are many of those. 

Review some typos and spelling. 

Author Response

Point-by-point responses to Reviewer comments

We thank the Reviewers for their criticisms and suggestions. The manuscript has been revised accordingly and additional points are discussed below. For the Reviewers’ convenience, the revisions have been highlighted in the manuscript.

Reviewer 1:

In this review by Moretti et al. the authors explored several aspects of HIV pathogenesis and the gut tissue, from cell reservoir and its interaction with the microbiome to HIV-associated comorbidities.

The review is well-structured and written. It gives the reader a broad understanding of a topic and provides up-to-date references if they want to explore particular topics.

One critical comment is about some terminology. Please avoid (and I suggest changing it) using "HIV-positives individuals", "HIV-infected people", "HIV-infected patients" and instead use PLWH.

I would also suggest changing a bit the colors in Figure 1. The green and the red are too dark and make it hard to see the cell label. Also, the cell name is very small, and it is a bit hard to differentiate Tcm from Tem. Maybe you can try to add a legend beside the figure rectangle for each cell type to make it "cleaner" and easier to look at.

A minor comment, check carefully for typos (example in line 33). Also, many words have a dash in the middle, for example in lines 35, 143, 183, and so on. It could have happened in the submission and formatting process, but it needs to be carefully revised since there are many of those.

Re: We thank the Reviewer for his/her positive assessment of our manuscript and the constructive comments. Critical comments about terminology and typos have been addressed. According to the Reviewer’s suggestion, Figure 1 has been modified and we hope it now meets with the Reviewer’s demand.

Reviewer 2 Report

Moretti et al examine the interactions of intestinal mucosa cells including CD4 T  cell subsets, the gut microbiome and inflammatory and immune responses in the pathogenesis of HIV infection.  Overall, this is very well-written and generally comprehensive review. There are a few areas of related research that, if included, would markedly strengthen the review. Discussion of important areas of future research is often an important feature of reviews such as this one, but such a discussion is relatively brief in this review (only a couple of interesting areas mentioned briefly). Figure 1 could also be revised to be more informative and in line with the text.

Specific Points:

  1. Figure 1 is worth modifying to better reflect statements in the text and to be more accurate and informative. Are the relevant cells in the right places (for example, the figure does not show Tfh in Peyer’s Patches but instead in the blood).  Similarly, itmight be helpful to show HIV infection of some of the additional cell types indicated in the text to be infected, but not shown in the figure (ex. Tregs).

Additional Discussion Points that would improve the mansucript

  1. The authors discuss the induction of fibrosis of lymphoid tissue with HIV progression. This raises the question about what is known about Peyer’s patches structure/organization/function and HIV infection.  A cursory look at the literature suggests that there maybe structural changes associated with HIV infection (Levesque PMID: 19582166) that may be relevant. 
  2. Understanding that the review is focused on gut-associated lymphoid tissue, it is clear that there are also important effects of HIV infection on gut associated myeloid-derived cells (DCs, macrophages) that are critical in the gut immune response. Some additional discussion of HIV infection ion these cells and how this may impact on the lymphoid cell infection and the microbial translocation, would provide a broader view of the cells involved in pathogenesis. This also appears to be an area for which there

Minor points:

There are some relatively minor English usage issues in terms of plural agreements and verb tense agreement. A couple are listed below, but this is not an all-inclusive list.

Line 69 should be “is lined by” (not lining by). Mucus layer should be plural layers, (not layer) 

line 73, I assume it should read: “different levels of defence mechanisms to limit translocation out of the lumen (rather than “into” the lumen)

Good. There are a few areas of non-optimal English usage (mostly agreements of plurals, and verbs) that should be adjusted during editing for final publication, but quite minor.

Author Response

Point-by-point responses to Reviewer comments

We thank the Reviewers for their criticisms and suggestions. The manuscript has been revised accordingly and additional points are discussed below. For the Reviewers’ convenience, the revisions have been highlighted in the manuscript.

Reviewer 2:

Moretti et al examine the interactions of intestinal mucosa cells including CD4 T cell subsets, the gut microbiome and inflammatory and immune responses in the pathogenesis of HIV infection.  Overall, this is very well-written and generally comprehensive review. There are a few areas of related research that, if included, would markedly strengthen the review. Discussion of important areas of future research is often an important feature of reviews such as this one, but such a discussion is relatively brief in this review (only a couple of interesting areas mentioned briefly). Figure 1 could also be revised to be more informative and in line with the text.

Specific Points:

Figure 1 is worth modifying to better reflect statements in the text and to be more accurate and informative. Are the relevant cells in the right places (for example, the figure does not show Tfh in Peyer’s Patches but instead in the blood).  Similarly, it might be helpful to show HIV infection of some of the additional cell types indicated in the text to be infected, but not shown in the figure (ex. Tregs).

Re: We thank the Reviewer for his/her careful and insightful review of our manuscript and the constructive comments. According to the Reviewer’s suggestion, Figure 1 has been modified and we hope it now meets with the Reviewer’s demand. Regarding the Reviewer's request on the “Discussion of important areas of future research…”, we implemented the “Conclusions” by adding some additional discussion on new strategies aimed at recovering the intestinal homeostasis that could benefit HIV‐1‐infected individuals, particularly those on different ART regimens.

Additional Discussion Points that would improve the manuscript

The authors discuss the induction of fibrosis of lymphoid tissue with HIV progression. This raises the question about what is known about Peyer’s patches structure/organization/function and HIV infection.  A cursory look at the literature suggests that there maybe structural changes associated with HIV infection (Levesque PMID: 19582166) that may be relevant.

Understanding that the review is focused on gut-associated lymphoid tissue, it is clear that there are also important effects of HIV infection on gut associated myeloid-derived cells (DCs, macrophages) that are critical in the gut immune response. Some additional discussion of HIV infection on these cells and how this may impact on the lymphoid cell infection and the microbial translocation, would provide a broader view of the cells involved in pathogenesis. This also appears to be an area for which there

Re: We thank the Reviewer for his/her suggestions that are relevant to improve the review. As requested by the Reviewer, all of the issues has been implemented accordingly. A more detailed description of the structural changes of Peyer’s patches structure/organization/function associated with HIV infection, and of the effects of HIV infection on gut associated myeloid-derived cells and how this may impact on the lymphoid cell infection and the microbial translocation have been included in the revised version of the manuscript.

Minor points:

There are some relatively minor English usage issues in terms of plural agreements and verb tense agreement. A couple are listed below, but this is not an all-inclusive list.

Line 69 should be “is lined by” (not lining by). Mucus layer should be plural layers, (not layer)

line 73, I assume it should read: “different levels of defence mechanisms to limit translocation out of the lumen (rather than “into” the lumen)

Comments on the Quality of English Language

Good. There are a few areas of non-optimal English usage (mostly agreements of plurals, and verbs) that should be adjusted during editing for final publication, but quite minor.

Re: According to the Reviewer’s suggestion, manuscript has been improved with the help of a native English speaking assistant.